# Sweet chestnut standardized fractions from sustainable circular process and green tea extract: *In vitro* inhibitory activity against phytopathogenic fungi for innovative applications in green agriculture

Annalisa Romani[1,2], Gabriele Simone[2,3], Margherita Campo [1,4]*, Lorenzo Moncini[3], Roberta Bernini [5]

1 Department of Statistics, Computing, Applications "Giuseppe Parenti" DiSIA, PHYTOLAB (Pharmaceutical, Cosmetic, Food supplement Technology and Analysis), University of Florence, Florence, Italy, 2 PIN scrl, "City of Prato" University Center-Educational and Scientific Services for the University of Florence, Prato, Italy, 3 Centro Ricerche Strumenti Biotecnici nel settore Agricolo-forestale (CRISBA), ISIS "Leopoldo II di Lorena", Grosseto, Italy, 4 Consortium I.N.S.T.M., Florence, Italy, 5 Department of Agriculture and Forest Sciences (DAFNE), University of Tuscia, Viterbo, Italy

* margherita.campo@unifi.it

**Data Availability Statement:** All relevant data are within the manuscript and figures.

## Abstract

In the present study, the antifungal activities of two commercial tannins-rich dry fractions towards different filamentous fungi of agronomical and food interest were evaluated. In particular, a standardized fraction from sweet chestnut (*Castanea sativa* Mill.) wood by-products and a commercial green tea (*Camellia sinensis* L.) leaf extract were tested at different concentrations (0.1–5.0% and 0.2% w/v respectively). The Sweet Chestnut Wood fraction was produced in an industrial plant through an environmentally and economically sustainable process, involving hot-water extraction and a sequence of membrane filtration steps with different molecular cut-offs for fractionation and concentration of the active principles. The Sweet Chestnut Wood and Green Tea Leaf extracts were characterised *via* HPLC/DAD/MS quali-quantitative analysis. The first extract showed a polyphenolic content of 20.5% w/w, 100% hydrolysable tannins; the second one showed a polyphenolic content of 87.5% w/w, of which 96.2% epigallocatechin gallate and 3.8% epicatechin gallate. The antifungal activity of the Sweet Chestnut fraction in aqueous solutions was evaluated towards different filamentous fungi, in particular telluric phytopathogens (*Fusarium oxysporum* f. sp. *radicis-lycopersici*; *Fusarium solani*; *Rhizoctonia solani*; *Sclerotium rolfsii*) and post harvest pathogens (*Botrytis cinerea*, that can also attack field plants; *Penicillium digitatum*; *Penicillium italicum*), and compared to the activity of Green Tea Leaf extract solutions. The experimental results evidenced, for almost all tested fungi, inhibition of the mycelial growth rate in presence of tannins. The lowest inhibitions were observed for *B. cinerea* (7.5%, to 28.9%) and *P. italicum* (53.8% in 5.0% w/v Sweet Chestnut extract substrate). A proportional inhibitory effect to tannin concentration was observed for *F. oxysporum* f. sp. *radicis-lycopersici* and *F. solani* (from 33.7% to 56.6%), *R. solani* (from

**Funding:** This research is part of the following projects: Tuscany Region, POR FESR 2014/2020. Nat-BackeryInnov "Innovative production of a bakery line based on natural functional extracts for wellness and sports"; Tuscany Region, POR CREO FESR 2014-2020 "Formulation of new sustainable products for agriculture based on natural extracts", LIFE18 ENV/NL/000043 PlantsforPlants: "Boost conventional agriculture's confidence: new organic bio-stimulants to reduce water, nutrients and pesticide demand".

**Competing interests:** The authors have declared that no competing interests exist.

29.7% to 68.8%) and *P. digitatum* (64.7% to 87.0%). The highest effect resulted for *S. rolfsii*, (5.0% to 100%).

## Introduction

In the recent years, the European legislation aimed to strongly reduce the impact of agricultural practices on environment and on health and wellness of operators and consumers. Community new Directives and Regulations, gradually implemented by specific Legislative Decrees at national level, imposed the revision of all criteria and rules for the production, market and use of active substances and products for plant protection in the European Union. This led to a progressive reduction of use and doses of traditional chemicals and products, thus limiting or prohibiting the diffusion of the most harmful substances, and actually discouraging the use of traditional pesticides in favour of sustainable agriculture practices with low environmental impact and minimal or negligible risk for the health of operators and users of the finished products. Thanks to the introduction of innovative and increasingly advanced technologies respectful of the environment, such as integrated defence systems and search for new natural and ecological solutions, agrochemicals consumption recently fell from 140,000 to 95,000 tons average per year. The progressive entry into force of the new regulations determined, for 2018, a negative trend for traditional chemicals, whereas organic agrochemicals are today worth about 20% of the market.

In agricultural systems, fungal phytopathogens are responsible for 70% of the total losses caused by pathogenic microorganisms [1]. The infections can occur either in field or in postharvest phase, inducing alterations in the development or death of plants and loss of products or of their organoleptic and nutritional properties. Moreover, some of them can be responsible for intoxications and allergic disorders due the production of mycotoxins and allergens. To counteract these pathogens, synthetic fungicides are necessary but the negative side effects of their use, including their toxicity, are relevant on both human health and environment. In consideration of the increasing attention devoted to develop sustainable and eco-friendly processes in crop production, plant extracts, mainly those particularly rich in tannins and other polyphenolic compounds, could represent a valid solution as an alternative to synthetic fungicides [2].

Tannins are a wide sub-class of naturally occurring compounds that belongs to the chemical class of polyphenols. The variability of their structures allows for a further distinction, mainly in hydrolysable and condensed tannins: the first are polyhydroxylated compounds, usually *D*-glucose, partially or totally esterified with phenolic acids such as gallic acid or ellagic acid to get, respectively, gallotannins and ellagitannins; condensed tannins are oligomers or polymers based on flavan-3-ol units, commonly catechin or epicatechin, linked *via* carbon-carbon bonds, in some cases esterified with gallic acid [3]. They play a role of defence for plants against pathogenic microorganisms and their action is based on both oxidative and toxicity mechanisms due to enzymatic inhibition, but also to the ability to selectively interact with lipids in bacterial membranes by modifying their properties and irreversibly damaging them. Since ancient times, tannins have been used as mordents in textiles, for tanning of leather and to clarify wines. Recent studies reported several biological properties that make them suitable also for more specific applications in agronomy, veterinary, cosmetics, food, medicine and phytotherapy [4–12]. Hydrolyzable tannins, in particular tannic acid, are known for their ability to induce beneficial effects on human health including anti-mutagenic, anticancer and

antioxidant activities. In addition, their ability to reduce serum cholesterol and triglycerides and to suppress lipogenesis by insulin has been well documented in the literature [13, 14]. The effect of hydrolysable tannins on membrane lipids was evaluated by studies on liposomes, whose results showed dose-dependent and time-dependent damage to the lipid bilayer by all of the tested compounds. A significant ability of gallic and ellagic derivatives to inhibit the synthesis of chitin, an essential component of the yeast cell wall, has been demonstrated; this inhibition could be one of the main factors that specifically determine the action against yeasts [15]. Condensed tannins show interesting antioxidant, anti-allergy, anti-hypertensive, and antimicrobial activities [16, 17]. In particular, several studies reported the antioxidant and antimicrobial properties of Green Tea (*Camellia sinensis* L.) Leaf (GTL) extracts, due to the high content of condensed tannins, mainly epigallocatechin gallate (EGCG). The antimicrobial effects of these extracts were demonstrated against gram-positive and gram-negative bacteria as *Escherichia coli*, *Salmonella* spp., *Staphylococcus aureus*, *Enterococcus* spp., fungi as *Candida albicans* and viruses (e.g., HIV, *Herpes simplex*, *influenza*) with Minimum Inhibitory Concentrations (MICs) in the range from 0.156 to 0.313 mg/mL of extract [18–21].

Plant extracts rich in hydrolysable tannins generally have more complex chemical compositions and they are also less stable with respect to condensed tannins rich extracts, so there are some objective difficulties both in the study of their composition and biological properties, and in obtaining industrial standardized and stable products to be marketed. Nowadays, the most of the scientific results on matrices rich in hydrolysable tannins concern chestnut, oak and pomegranate. Among them, sweet chestnut (*Castanea sativa* Mill.) extracts are the most water-soluble, well industrially standardized and available at low costs.

In the present study, the *in vitro* inhibitory effect of one industrial, standardized and sustainable natural fraction from Sweet Chestnut Wood (SCW) was assessed against selected phytopathogenic fungi of agronomical interest, and compared to the activity of one registered commercial GTL extract, at different concentrations. The SCW fraction under study is produced by applying an industrial circular and sustainable process whose products were previously optimized and chemically characterized for their content in hydrolysable tannins, studied for their specific antioxidant properties, and compared with pomegranate, myrtle and other natural extracts, also containing hydrolysable tannins or different subclasses of polyphenols [22–24]. In particular, the above mentioned SCW fraction and GTL extract were tested against telluric phytopathogens (*Fusarium oxysporum* f. sp. *radicis-lycopersici*; *Fusarium solani*; *Rhizoctonia solani*; *Sclerotium rolfsii*) and post-harvest pathogens (*Botrytis cinerea*, that can also attack field plants; *Penicillium digitatum*; *Penicillium italicum*). All these fungi can grow either as saprophytic, consuming the decaying matter, or as pathogens. *Fusarium*, *Rhizoctonia* and *Sclerotium* can attack a wide range of cultivated herbaceous plants causing various damages, like roots rot and the damping-off of seedlings and adult plants. Moreover, genus such as *Sclerotium* can produce sclerotia that can enable them to survive on harsh conditions on the soil and re-infect the crop year by year. *B. cinerea* is a pathogen that can attack either plants in soil and their products in post-harvest phase. The main host are grapevine, but they can attack also strawberries, tomatoes and other plants, striking fruits and sometimes also stems and leaves. Finally, *P. digitatum* and *P. italicum* are mainly known as pathogens of fruits in post-harvest phase, which cause serious losses annually. An infected lesion starts to be saggy and watery, then the mycelium emerges and starts to sporulate. The spores can be dispersed extremely easily and can re-infect the material in the nearby. To the best of our knowledge, tannin-rich extracts were investigated for their effects against bacteria and yeasts, but few studies are available on their effect against filamentous fungi, whereas the same industrial SCW fraction was previously studied [22], compared with other polyphenols-rich extracts, against pathogenic bacteria of agricultural interest such as *Pseudomonas savastanoi* and *Pseudomonas*

*syringae* by both *in vitro* tests and on economically relevant species such as olive tree, kiwi, tomato and carrot, within the activities of the European Project LIFE EVERGREEN, "Environmentally friendly biomolecules from agricultural wastes as substitutes of pesticides for plant diseases control", LIFE13 ENV/IT/000461, concluded in 2016. The activities are still carrying on in the new LIFE Project Plants for Plants, "Boost conventional agriculture's confidence: new organic bio-stimulants to reduce water, nutrients and pesticide demand", LIFE18 ENV/NL/000043.

In this study, both the SCW dry fraction and GTL extract were chemically characterized by HPLC/DAD/MS for their contents in polyphenolic secondary metabolites, before testing them against the above mentioned phytopathogens. The parallel tests with the commercial GTL extract, rich in condensed tannins, EGCG in particular as demonstrated by HPLC/DAD/MS analysis, allowed for comparing the effects of the SCW fraction to those of a widely investigated product whose antimicrobial properties are exploited for different purposes such as textiles antimicrobial treatments and food safety and quality. The presence of gallic acid units, in both hydrolysable tannins from SCW fraction and EGCG from GTL extract, is believed to play an important role in the mechanism of antimicrobial activity [25, 26].

This study is based on the above reported previous findings and considerations, as a prosecution of the research activities on sustainable products rich in natural compounds. It is aimed at an extension of the possible applications of this natural, sustainable industrial fraction to further targets in a productive sector like the agro-industry, where there is a large demand of new and innovative standardized, industrialised products economically and environmentally sustainable. The natural SCW fraction under study could represent a valid innovation in the market of agro-pharmaceuticals in general, also considering that nowadays several natural products and preparations are commonly developed and marketed in a limited context or as non-standardized formulations, often not competitive at an industrial scale. This study could represent a first step for the development of standardised and industrialised ecological crop protection products, suitable for use in green agriculture, in compliance with the regulations and according to environmental sustainability and health safety.

## Materials and methods

### Chemicals

All solvents (HPLC grade), formic acid (ACS reagent) and EGCG were purchased from Sigma Aldrich Chemical Company Inc. (Milwaukee, Wisconsin, USA). Gallic and ellagic acids, of analytical grade, were purchased from Extrasynthèse S.A. (Lyon, Nord-Genay, France). HPLC-grade water was obtained via double-distillation and purification with a Labconco Water Pro PS polishing station (Labconco Corporation, Kansas City, USA).

### Extracts

The SWC dry extract was a commercial fraction furnished by GRUPPO MAURO SAVIOLA Srl (Viadana, MN, Italy). This fraction is obtained after ten circular and solvent free process streams by the industrial tannin extraction and concentration/purification plant operating in Radicofani (SI, Italy), previously described [23, 27]. Briefly, the ten process streams are: (1) filtration of tannin broths; (2) permeation from nanofiltration step-1; (3) concentration from nanofiltration step-1; (4) concentration from nanofiltration step-1 (after cooling); (5) permeation from nanofiltration step-2; (6) concentration from nanofiltration step-2; (7) osmosis permeation; (8) osmosis concentration; (9) settled fraction from clarification step; and (10) spray-dried material obtained from fraction 6 (commercial SWC dry extract under examination).

The GTL was the commercial dry extract known as TEAVIGO® (DSM Nutritional Products, Heerlen, Netherlands).

## Phytopathogens

The phytopathogens used for this work and their source are shown in Table 1. All fungi, in collection at the CRISBA research center (Grosseto, Italy) where the *in vitro* tests are performed, were cultivated on PDA (Formedium ltd., Norfolk, UK) at 25°C and, when necessary, kept at 4°C.

## HPLC/DAD/ESI-MS analysis

SWC fraction and GTL extract were analyzed by using a HP-1200 liquid chromatograph equipped with a DAD detector and a HP 1100 MSD API-electrospray (Agilent Technologies, Santa Clara, CA) operating in negative and positive ionization mode. A Luna, C18 250×4.60 mm, 5 μm column (Phenomenex, Torrance, CA), operating at 26 °C was used. The eluents were $H_2O$ (adjusted to pH 3.2 with HCOOH) and $CH_3CN$. A four-step linear solvent gradient starting from 100% $H_2O$ up to 100% $CH_3CN$ was performed with a flow rate of 0.8 mL/min over a 55 minutes period, as previously described [23, 27]. Mass spectrometer operating conditions were: gas temperature 350 °C at a flow rate of 10.0 L/min, nebulizer pressure 30 psi, quadrupole temperature 30 °C and capillary voltage 3500 V. The fragmentor was set at 120 eV.

## Quali-quantitative analysis

The polyphenols present in the extracts were identified by using data from HPLC/DAD and HPLC/MS analyses, by comparing their retention times, UV-Vis and mass spectra with those of the available specific commercial standards. Each compound was quantified by HPLC/DAD, using a five-point regression curve built with the available standards. Calibration curves with $r^2 \geq 0.9998$ were considered. The concentration of each compound was calculated by applying the appropriate corrections for changes in molecular weight. Gallic acid was calibrated at 280 nm and ellagic acid at 254 nm with the appropriate standards; EGCG and epicatechin gallate (ECG) were calibrated at 280 nm using EGCG as reference. The evaluation of the polyphenol content was carried out in triplicate. The results were recorded as mean values with standard deviations ≤ 5%. The polyphenolic extracts were tested at tannin concentrations

**Table 1. Phytopathogens used and their original source.**

| Fungal strain | Source |
|---|---|
| *Botrytis cinerea* SAS405 | Strawberry (Italy) [a] |
| *Fusarium oxysporum* f. sp. *radicis-lycopersici 8627* | Tomato (Italy) [a] |
| *Fusarium solani* 10798 | Passionflower (Italy) [a] |
| *Rhizoctonia solani* RB4 | Tobacco (Italy) [a] |
| *Sclerotium rolfsii* 10827 | Tomato (Italy) [a] |
| *Penicillium digitatum* CECT20796 | Rotten fruit (Spain) [b] |
| *Penicillium italicum* FVP10 | Tangerine (Italy) [c] |

[a] Mycological Laboratory of Plant Pathology of the Agricultural Science, Alimentary and Agro-environmental Department, University of Pisa, Italy.

[b] Spanish Type Culture Collection (CECT), University of Valencia, Spain.

[c] Department of Soil, Plant and Food Science, University of Bari "Aldo Moro", Italy.

between 0.364 mM and 18.200 mM (SCW) and 3.828 mM (GTL), calculated according to the reported HPLC/DAD analysis results.

## Antifungal screening

*In vitro* **test of radial growth inhibition.**    The inhibitory effect of extracts on fungal growth were tested using the "poison food technique method", according to the literature [28]. The SCW fraction was added to the PDA at the final concentrations of 1.0%, 2.0% and 5.0% w/v, the pH was adjusted at 7.0 adding KOH 10 M, autoclaved and then poured in 85 mm Petri dishes. A control (PDA without the extracts) was used. For fungi that were totally inhibited at these concentrations, two additional media were prepared with SCW at 0.1% and 0.5% w/v for other tests. The plates were inoculated at the center with 5 mm mycelium discs cut from the 7-day-old cultures.

Otherwise, *P. italicum* was inoculated using a conidial suspension according to the procedure described by Kinay *et al.*, 2006 with some modifications [29]. Briefly, the conidial suspension was prepared scraping the conidia from a 7-day-old colony and diluting it in distilled water to reach $10^6$ conidia/mL; 50 μL of this suspension were placed in a 5 mm hole made in the center of the solidified medium. The plates were then incubated at 25˚C in darkness and colonies diameters were recorded every 24 h, until the reaching of the Petri dish's border. The diameters values were the mean of three replicates for each thesis. The percentage of inhibition value was calculated at the last interval of time, according to the following equation [30].

$$I = [(CFc - CFt)/CFc] \ x \ 100$$

where I = inhibition (%); CFc = control growth (diameter); CFt = treatment growth (diameter).

**Comparison test between SCW fraction and GTL extract.**    To compare the effects of SWC fraction and GTL extract, the previous test was repeated using the extract at 0% (control) and 2% and a media prepared with GTL at 0.2% w/v. The preparation of the media, the inoculation and the data collection were carried out as described above (see "*In vitro* test of radial growth inhibition").

**Sclerotia germination inhibition test.**    The aim of this test is to evaluate the inhibitory property of the SCW extract on sclerotia germination. Three media were prepared as previously described, adding SCW extract to the PDA at the final concentrations of 0% (control), 0.5% and 2.0% w/v. The mature sclerotia were obtained from a 45-day-old colony grown on PDA and dried at 30˚C for 20 h to induce the eruptive germination. The sclerotia were sterilized with NaClO (2% active chlorine) for 3 minutes and washed in distilled sterile water for 3 times [31]. Each Petri dish was inoculated with 20 sclerotia and 5 replicates were made for each treatment (a total of 100 sclerotia for each thesis). After 5 days, the number of germinated sclerotia was registered and the sclerotial germination inhibition rate was calculated according to the following equation [31]:

$$IR_{SG} = [(NCK - Nt)/NCK] \ x \ 100$$

where: $IR_{SG}$ is the sclerotial germination inhibition rate (%);
 NCK = germination rate on control thesis;
 Nt = germination rate on treated control.

## Statistical analyses

Data were elaborated using DSAASTAT program through the variance analysis (ANOVA). The means separation was carried out using the Duncan's H.M. multiple range test prior

angular transformation for the percentage values. The statistical significance was evaluated at $p < 0.01$ for each test and at $p < 0.05$ for the sclerotial germination test [32].

## Results and discussion

### Quali-quantitative HPLC/DAD/ESI-MS characterization of SCW fraction and GTL extract

Both SCW fraction and GTL extract were commercially available. The SWC fraction was obtained *via* aqueous extraction and membranes technology purification and concentration from SWC wood by green methodologies. The industrial plant for hydrolysable tannins extraction, previously described [23, 27], was designed for the economically and environmentally sustainable recovery of wood chips yielded as a by-product during the processing of SWC wood. The raw extract, obtained through hot water extraction, was purified and concentrated by a sequence of filtration steps on membranes with different molecular cut-offs, to obtain different fractions enriched in hydrolysable tannins with specific biological properties. Two of these industrial fractions are commercial: one liquid purified and concentrated fraction and the spray-dried powder obtained from this latter. The latter was the fraction investigated in the present study, and selected for its better characteristics of chemical and biological stability and industrial standardization. GTL extract was also commercially available as TEAVIGO® (see "Materials and methods" section).

SCW fraction and GTL extract were analyzed by HPLC/DAD/ESI-MS; the results are reported in Table 2. According to data already reported by Campo et al. [23], the polyphenols content in the SCW fraction consists entirely of hydrolysable tannins. With respect to the

**Table 2. Quali-quantitative HPLC/DAD/MS analysis of SCW fraction and GTL extract.**

| SCW fraction | mg/g[a] | mmol/g[a] |
|---|---|---|
| Vescalin | 9.284 ± 0.322 | 0.015 ± 0.0007 |
| Castalin | 8.633 ± 0.371 | 0.014 ± 0.0007 |
| Pedunculagin I | 9.964 ± 0.401 | 0.013 ± 0.0006 |
| Monogalloyl glucose I | 5.321 ± 0.098 | 0.016 ± 0.0004 |
| Gallic acid | 14.537 ± 0.281 | 0.086 ± 0.0018 |
| Monogalloyl glucose II | 4.961 ± 0.213 | 0.015 ± 0.0006 |
| Roburin D | 5.817 ±0.194 | 0.003 ± 0.0001 |
| Vescalagin | 47.379 ± 1.605 | 0.051 ± 0.0018 |
| Dehydrated tergallagic-*C*-glucoside | 2.091 ± 0.094 | 0.003 ± 0.0002 |
| Castalagin | 40.713 ± 1.378 | 0.044 ± 0.0016 |
| Digalloyl glucose | 3.698 ± 0.075 | 0.008 ± 0.0002 |
| *O*-Galloyl-castalagin isomer | 17.002 ± 0.376 | 0.031 ± 0.0007 |
| Trigalloyl glucose | 12.658 ± 0.382 | 0.020 ± 0.0006 |
| Tetragalloyl glucose | 9.996 ± 0.246 | 0.013 ± 0.0003 |
| Ellagic acid | 8.747 ± 0.732 | 0.029 ± 0.0025 |
| Pentagalloyl glucose | 4.637 ± 0.298 | 0.005 ± 0.0003 |
| *Total tannins* | *205.438 ± 7.380* | *0.364 ± 0.0130* |
| GTL extract | mg/g | mmol/g |
| EGCG | 841.610 ± 18.211 | 1.838 ± 0.040 |
| ECG | 33.698 ± 0.671 | 0.076 ± 0.002 |
| *Total tannins* | *875.308 ± 18.882* | *1.914 ± 0.042* |

[a]Data are expressed as mg and mmol of compound per gram of dry extract.

**Fig 1. Chemical structures of the main polyphenols found in SCW fraction and GTL extract.**

studies concerning the characterization of the wood matrix of sweet chestnut, the heaviest molecules and oligomers are not present in consideration of the hydrolysis process of their partial or total amounts during the extraction procedure in hot water, but the representative compounds of the vegetal species under study, as vescalagin and castalagin, are mostly preserved. Their amounts in the tested sample are, respectively, 23.1% and 19.8% w/w (on total tannins); small amounts of their partial hydrolysis products as vescalin, castalin and peduncu-lagin I were detected (4.5%, 4.2% and 4.9% w/w on total tannins, respectively). The dried SCW fraction was chemically stable as demonstrated by control HPLC/DAD analysis repeated after 6 and 12 months.

GTL extract showed a tannins content of 875.308 mg/g, 96.2% EGCG and 3.8% epicatechin gallate (ECG) (% w/w individual compounds on total tannins). Thus, as expected, it contains only monomeric condensed tannins derived from galloylated flavan-3-olic units, chemically more stable than hydrolysable tannins (Table 2). EGCG has a structural analogy with hydroly-sable tannins found in SCW fraction, represented by the presence of the gallic acid unit, which is believed to play an important role in the mechanism of antimicrobial activity (Fig 1).

### *In vitro* antifungal screening

Table 3 shows the tannins concentrations, expressed as mM, for each one of the tested solu-tions. They were calculated based on the weight of dry extract in aqueous solution and the

**Table 3. Tannins concentrations (mM) in the tested solutions, according to the % w/v of extracts and the quali-quantitative HPLC/DAD/MS data.**

| Tested concentration of extract (w/v) | 0.1% | 0.2% | 0.5% | 1.0% | 2.0% | 5.0% |
|---|---|---|---|---|---|---|
| Tannins in SCW solutions (mM) | 0.364 | | 1.82 | 3.64 | 7.28 | 18.2 |
| Tannins in GTL solution (mM) | | 3.83 | | | | |

chemical quali-quantitative characterization data shown in Table 2, in particular using the results expressed as mmol of total tannins in the extract. According to the results, the mM concentration of EGCG and ECG in the tested solution of 0.2% w/v GTL extract (3.83 mM) is similar to that of hydrolysable tannins in the 1.0% w/v SCW solution (3.64 mM); thus, the most direct comparison is between the results obtained for these two samples. Fig 2 shows the images of mycelial growth for the investigated fungi in the presence of different SCW concentrations. The experiment data of antifungal tests showed that both extracts inhibited the mycelial growth rate for almost all tested fungi. The results are shown in Fig 3 as growth inhibition (%) compared to the control. The lower growth rate inhibition was observed for *B. cinerea* (7.5%, 8.1% and 28.9% with SCW fraction at 1.0%, 2.0% and 5.0% w/v, respectively) and *P. italicum*, that was inhibited only with 5.0% w/v of SCW fraction (53.8% of inhibition), while it was slightly stimulated on lower doses. A proportional inhibitory effect to SCW concentration was observed for *F. oxysporum* f. sp. *radicis-lycopersici*, *F. solani* (from 33.7% to 56.6%), *R. solani* (from 29.7% to 68.8%) and *P. digitatum* (from 64.7% to 87.0%). The best result was obtained with *S. rolfsii*, that was totally inhibited with all concentrations of the extracts, so more tests were carried out at lower doses. A total inhibition of mycelial growth of *S. rolfsii* was confirmed with SCW fraction at 1.0% and 2.0% w/v, while an inhibition of the 86% on SCW 0.5% w/v (despite the high variability between replicas) and only 5.0% on SCW 0.1% was observed. In the literature, similar results were reported by testing gallic and ellagic acid against *B. cinerea* [33], *Fusarium* spp. [34] and *R. solani* [35].

## Comparison test between SCW fraction and GTL extract

Since the encouraging results obtained with the SCW fraction, we compared the effect of the 1% w/v concentration of this fraction with a comparable amount of GTL extract. In the literature, the effects of GTL extract against bacteria, yeast and some filamentous fungi are reported, but data about fungi of agronomic interest are lacking.

The results previously obtained with SCW fraction were confirmed and showed a better efficacy in some cases compared to GTL extract, as shown in Table 4. *P. italicum* was not inhibited by both the extracts. *B. cinerea* was scarcely inhibited (25.7%) but the results were better if compared to the SCW extract (7.5%). The inhibition of *F. oxysporum* f. sp. *radicis-lycopersici* and *F. solani* was of 13.9% and 25.9%, that was inferior compared to the 33.7% and 34.2% on tannins enriched medium. Better results were obtained against *R. solani* and *P. digitatum*: the inhibition with green tea and tannins was, respectively, 65% and 29.8% for *R. solani* and 65.2% and 64.8% for *P. digitatum*. In the latter case, the differences between the treatments are negligible and statistically not significant. Against *S. rolfsii* the inhibition (100%) was much higher for SCW fraction compared to GTL extract (46.6%). Some authors reported similar results against *Botrytis* with comparable concentrations of polyphenolic extract of GTL [36], while, at the best of our knowledge, literature data about the GTL extract against other fungi tested in this work are missing. The results show a better efficacy of SCW extract compared with GTL extract against the *Fusarium* and *Sclerotium*. A similar effect was observed against *P. digitatum*.

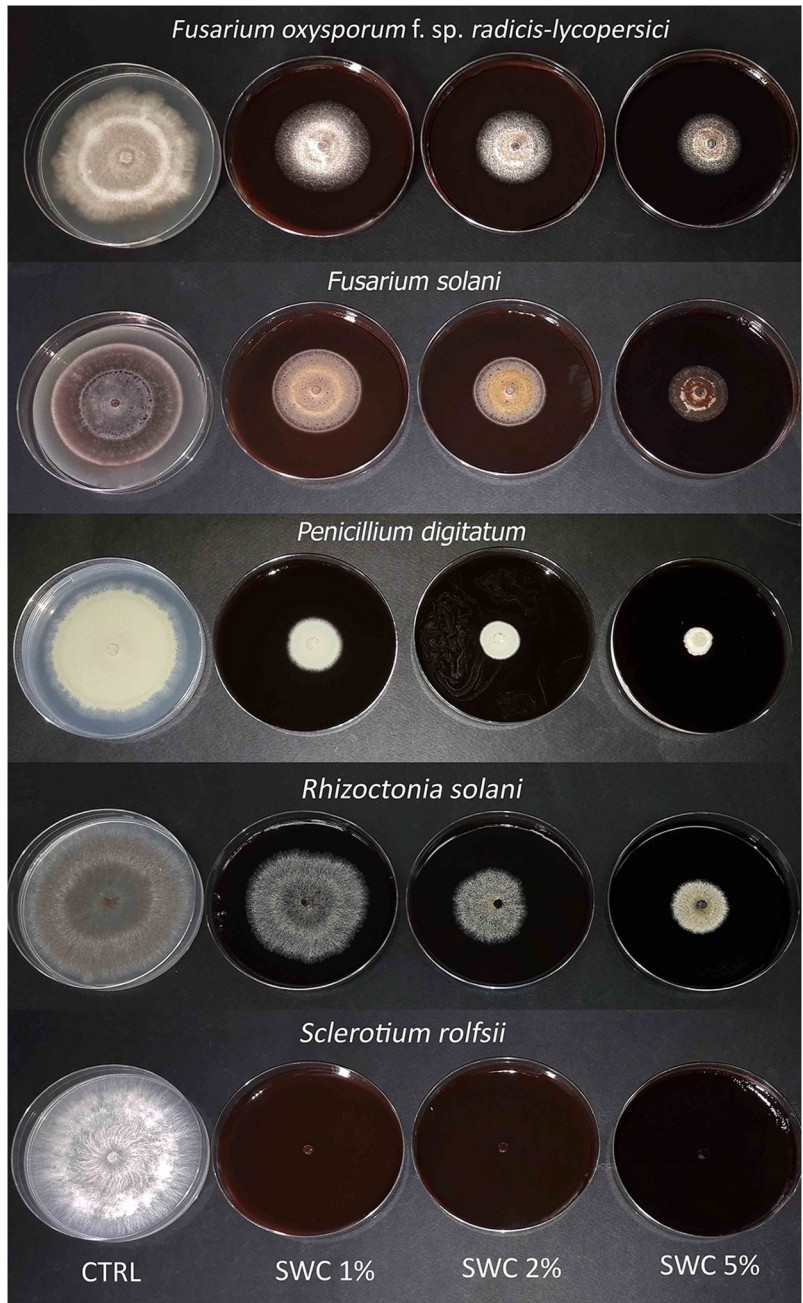

**Fig 2. Mycelial growth of the investigated fungi on the different SCW thesis.**

## Sclerotia inhibition test

In consideration of the effects of SCW fraction against *S. rolfsii*, additional tests were performed on the sclerotial germination. These tests were carried out using the lowest SCW concentration that inhibited the fungal growth (0.5% w/v) and the mean concentration used in this work (2.0% w/w). After 5 days of treatment with SCW fraction at 0.5% and 2.0% w/w, the sclerotial germination inhibition rate resulted, respectively, of 45.5% and 100%. At the end, the

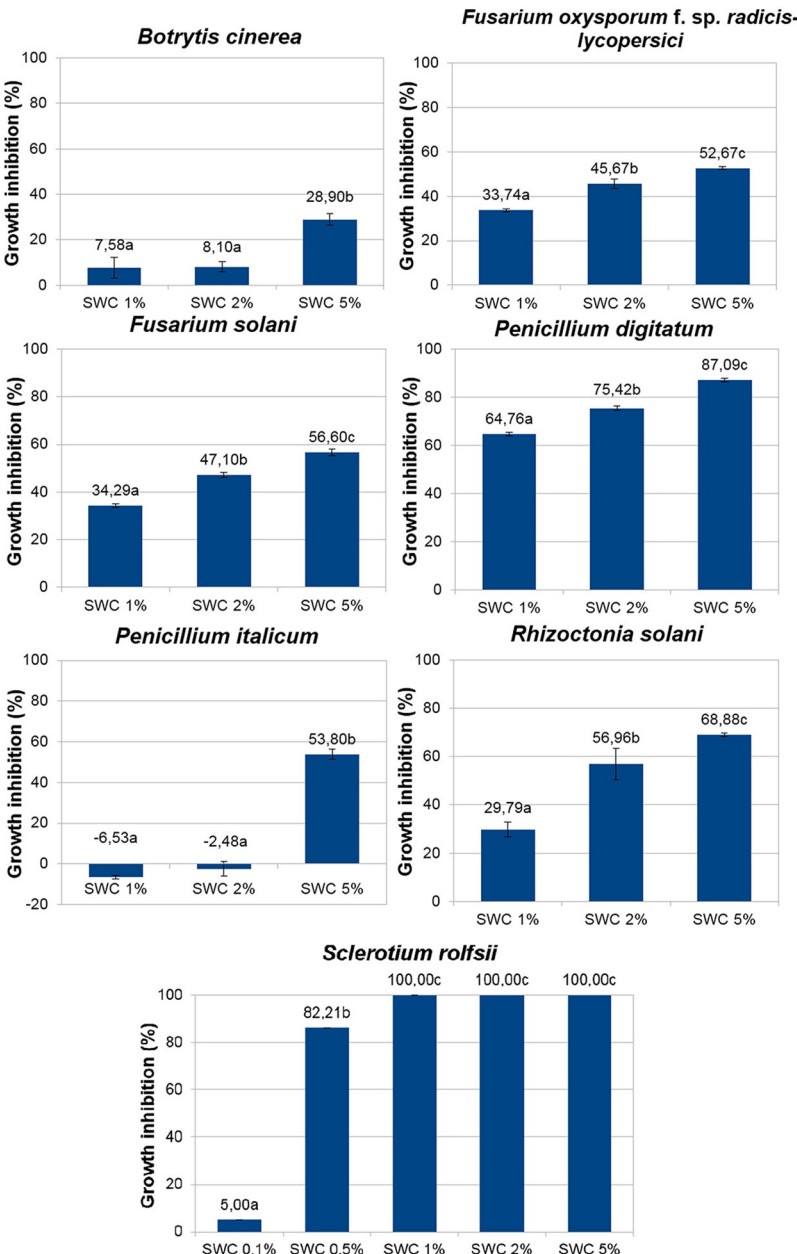

**Fig 3. Growth inhibition percentage for each tested phytopathogenic fungi.**

mycelium covered the entire plate on control, the colonies with the SCW fraction 0.5% w/w showed a maximum of 1 cm of diameter and no growth was observed with SCW 2.0% w/w. This implies that although the sclerotia can germinate, even if in lower percentage, the mycelium grows with difficulties. These results could be interesting to be applied in products to treat sclerotia-infested soil and prevent or reduce further infections. More studies will be carried out to explore this opportunity.

According to the activities and data above reported, this research fits into the context of the recent scientific studies, conducted at both basic and applicative research level, on the chemical characterization and biological properties of natural extracts and semi-finished products in

**Table 4. Inhibitory effect (%) on mycelial growth of the tested pathogens***.

|  | SCW extract (1% w/v) | GTL extract (0.2% w/v) |
|---|---|---|
| *Botrytis cinerea* | 7.5a | 25.7b |
| *Fusarium oxysporum* f. sp. *radicis-lycopersici* | 33.7a | 13.9b |
| *Fusarium solani* | 34.2° | 25.9b |
| *Penicillium italicum* | -6.5° | 3.3a |
| *Penicillium digitatum* | 64.8° | 65.2a |
| *Rhizoctonia solani* | 29.8° | 65.0b |
| *Sclerotium rolfsii* | 100c | 46.6b |

* Different letters in each row indicate a significant statistical difference (p<0.01).

order to standardize and industrialize green processes for the sustainable production of innovative references for applications in different sectors, to replace chemical synthetic active ingredients and additives.

Only recently the antimicrobial activity of hydrolysable tannins was explored with interesting results, but most of the scientific papers available concern the food safety sector. The results of these studies, some of which are in an applicative phase with industrial natural extracts, have shown the possibility of using hydrolyzable tannins from chestnut and other vegetal species as a potential substitute of synthetic food preservatives due to their antioxidant and antimicrobial properties [37–41]. These innovative applications are in line with the request from the market for natural and sustainable products, as a consequence of the reduction or elimination of traditional chemicals.

The same hydrolysable tannins fraction under study was previously tested [22], compared with other polyphenols-rich extracts, against pathogenic bacteria of agricultural interest such as *P. savastanoi* and *P. syringae* both *in vitro* and *in vivo* on economically relevant crops such as olive tree, kiwi, tomato and carrot. The work, conducted in collaboration with different research partners including university research centers and private companies, was mainly in the scope of the European Project LIFE—EVERGREEN. These results, together with the ones obtained for natural extracts rich in galloylated condensed tannins, constitute a context and a basis for this work.

According to the above reported results, interesting progresses are conceivable for some of the tested microorganisms, e.g. *P. digitatum* and *S. rolfsii*, as the percentage inhibitory effect of SCW fraction on mycelial growth is not only high from itself, but also equal or much higher than such found for GTL extract. Further studies are in progress to assess the *in vitro* and *in vivo* activity of SCW fraction towards other microorganisms and its stability under different conditions, to evaluate possible formulations for the finished products and eventual synergistic effects with other natural active extracts. In our opinion, this is the first report on the possibility of application of SCW extracts and natural active fractions in the green agricultural reality.

In order to verify and consolidate the reported results, further studies are in progress *in vitro* with similar pathogens and *in vivo* on economically relevant species such as olive tree, kiwi, tomato, basil and vitis. For the practical applications, the lower and medium doses are the most realistic in terms of application for practical reasons as costs and mixing operations. Compared to the GTL extract, the SCW fraction efficacy is higher or at least comparable in most of the analysed cases, but the sustainability of the industrial process and the lower costs, that make it affordable for large scale productions, are the main factors able to assess the SCW fraction as a valid natural antimicrobial for green agriculture together with GTL extract.

## Conclusion

This paper describes for the first time the activity of SCW fraction and GTL extract on pathogenic fungi of agronomic interest. While some results are anyway promising (as for *P. digitatum* and *S. rolfsii*), others could be the basis of further studies to investigate an eventual synergistic effect with other natural extracts with different action mechanisms. Relevant are the results for SCW fraction by a sustainable and reproducible industrial process using by-products of wood processing and environmental management of chestnut forests. The low cost of the raw materials and the efficiency of the process allow for obtaining a final product in large amounts, at a low final price, making it suitable for use in agriculture. Moreover, the compatibility with food uses permits an eventual application as a food preservative due to the antimicrobial and antioxidant activities. Further *in vitro* and *in vivo* studies are in progress to explore the effective potentiality of SCW fraction as an alternative to the use of synthetic fungicides in agriculture. A preliminary Life Cycle Assessment (LCA) study was performed, in comply with ISO 14040 and ISO 14044, for the quantitative assessment of the environmental, economic and social impacts deriving from the productive process of the commercial fractions, mainly related to soil and crops. The study was performed according to the reported extraction and purification/concentration methods of active compounds from sweet chestnut by the industrial extraction plant, consisting of a hot water-extraction system coupled with a membrane separation technology system, followed by the spray drying process to obtain the two commercially available finished products. The other fractions are reintroduced into the process to be further refined or added to the extraction water. They were taken into account parameters such as: climate change; ozone depletion; human toxicity; non-cancer effects; freshwater eutrophication; freshwater ecotoxicity; land use; water resource depletion; mineral and fossil resource depletion; loss of global species diversity and their functional traits as biodiversity parameters. The main benefits produced by using the investigated SCW fraction are related to all parameters evaluated, despite the large consumption of water for polyphenol extraction, thanks to the possibility of reusing water after filtration and purification treatments. These LCA results relate, in particular, the sustainability of the circular production process for obtaining sweet chestnut tannins. A further development of the LCA study, for the use of natural extracts in green agriculture, will include a specific evaluation of the economic and environmental sustainability, non-human toxicity and non-cancer effects, in replacing synthetic chemicals in green agriculture with antimicrobials and antioxidants based on natural active ingredients.

## Acknowledgments

The authors are grateful to GRUPPO MAURO SAVIOLA Srl (Viadana, MN, Italy) and Natural-mente Srl (Florence, Italy).

## Author Contributions

**Conceptualization:** Annalisa Romani, Margherita Campo, Roberta Bernini.

**Data curation:** Gabriele Simone, Margherita Campo.

**Formal analysis:** Gabriele Simone, Margherita Campo.

**Funding acquisition:** Annalisa Romani, Roberta Bernini.

**Investigation:** Gabriele Simone, Margherita Campo.

**Methodology:** Annalisa Romani, Gabriele Simone, Margherita Campo, Lorenzo Moncini, Roberta Bernini.

**Project administration:** Annalisa Romani, Roberta Bernini.

**Resources:** Annalisa Romani, Lorenzo Moncini, Roberta Bernini.

**Supervision:** Annalisa Romani, Margherita Campo, Roberta Bernini.

**Validation:** Annalisa Romani, Lorenzo Moncini, Roberta Bernini.

**Visualization:** Gabriele Simone, Margherita Campo, Lorenzo Moncini.

**Writing – original draft:** Annalisa Romani, Gabriele Simone, Margherita Campo, Lorenzo Moncini, Roberta Bernini.

**Writing – review & editing:** Annalisa Romani, Gabriele Simone, Margherita Campo, Roberta Bernini.

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
