## [Decision Letter · Decision Letter 0]

29 Jul 2020

PONE-D-20-10008

Sweet chestnut (Castanea sativa Mill.) standardized fractions from sustainable circular process and green tea (Camellia sinensis L.) extract: in vitro inhibitory activity against phytopathogenic fungi for innovative applications in green agriculture

PLOS ONE

Dear Dr. Campo,

Thank you for submitting your manuscript to PLOS ONE. After careful consideration, we feel that it has merit but does not fully meet PLOS ONE’s publication criteria as it currently stands. Therefore, we invite you to submit a revised version of the manuscript that addresses the points raised during the review process.

Your MS needs significant corrections before we consider this MS for further review for publication in PLOS One. Kindly do the needful changes as suggested by the reviewers and submit a point wise author response. 

We look forward to receiving your revised manuscript.

Kind regards,

Vijai Gupta, PhD in Microbiology

Academic Editor

PLOS ONE

Journal Requirements:

2.We suggest you thoroughly copyedit your manuscript for language usage, spelling, and grammar. If you do not know anyone who can help you do this, you may wish to consider employing a professional scientific editing service.  

3.Thank you for stating the following in the Financial Disclosure section:

[This research is part of the following projects:  Tuscany Region, POR FESR 2014/2020. Nat-BackeryInnov "Innovative production of a bakery line based on natural functional extracts for wellness and sports”; Tuscany Region, POR CREO FESR 2014-2020 “Formulation of new sustainable products for agriculture based on natural extracts”, LIFE18 ENV/NL/000043 - LIFE “PlantsforPlants”.].   

We note that one or more of the authors are employed by a commercial company: PIN scrl

Additional Editor Comments (if provided):

Dear Author, Your MS "Sweet chestnut (Castanea sativa Mill.) standardized fractions from sustainable circular process and green tea (Camellia sinensis L.) extract: in vitro inhibitory activity against phytopathogenic fungi for innovative applications in green agriculture" needs significant corrections before we consider this MS for further review for publication in PLOS One. Kindly do the needful changes as suggested by the reviewers and submit a point wise author response.

Reviewers' comments:

Reviewer's Responses to Questions

**Comments to the Author**

1. Is the manuscript technically sound, and do the data support the conclusions?

Reviewer #1: Yes

Reviewer #2: Yes

2. Has the statistical analysis been performed appropriately and rigorously? 

Reviewer #1: Yes

Reviewer #2: Yes

3. Have the authors made all data underlying the findings in their manuscript fully available?

Reviewer #1: Yes

Reviewer #2: Yes

4. Is the manuscript presented in an intelligible fashion and written in standard English?

Reviewer #1: Yes

Reviewer #2: Yes

5. Review Comments to the Author

Reviewer #1: This paper needs to be carefully reorganized with a clearly stated argument spelled out in the opening pages. The authors should work harder on the approach adopted, establish a clear theoretical background to contextualize the analysis and narrow the scope of the analysis to specific aspects. Apart from compiling key studies, a critical approach to the literature is required. There is a need of offering detailed insights regarding the main purpose of the paper and its contribution. The discussions require more structure and there is a need of offering a clear assessment of reviewed literature. The main contributions of the paper should be presented as part of the empirical discussions or critical assessment on the core research outcomes. The main contribution should be emphasised more and the concluding statements should be stronger. The paper presents its points in a rather descriptive manner, referring to relevant sources but without really discussing them. It brings up interesting points, but stating them rather than arguing for them. The research idea is not properly contextualised, as there is a need of offering a detailed review of relevant literature that help the authors developing the key arguments that support their proposed research. Data gathering and data analysis can be reconsidered and discussed more comprehensively. The lead up argument for the study and results section appear to be too loosely constructed. There is some discussion of the limitations of the study however these are not considered in terms of the implications on the study findings. Some bibliographic references are simply brought up without being developed, or without an adequate explanation as to why they are relevant.

Reviewer #2: Reviewer’s Report: PONE-D-20-10008

Sweet chestnut (Castanea sativa Mill.) standardized fractions from sustainable circular process and green tea (Camellia sinensis L.) extract: in vitro inhibitory activity against phytopathogenic fungi for innovative applications in green agriculture

Here are some minor corrections:

Line 316-318 figures title should be removed

Line 353 space should be removed

I suggest Accept the manuscript for publication in PLOS ONE.

6. PLOS authors have the option to publish the peer review history of their article (what does this mean?). If published, this will include your full peer review and any attached files.

Reviewer #1: No

Reviewer #2: No

---

## [Author Response · Author response to Decision Letter 0]

30 Sep 2020

We would like to thank the Editors for their comments. We checked PLOS ONE style templates and ensured that our manuscript meets the style requirements.

2.We suggest you thoroughly copyedit your manuscript for language usage, spelling, and grammar. If you do not know anyone who can help you do this, you may wish to consider employing a professional scientific editing service.

We had the manuscript checked for language usage, spelling, and grammar and fixed some inaccuracies, as evidenced in the text of our Manuscript.

3.Thank you for stating the following in the Financial Disclosure section:

[This research is part of the following projects: Tuscany Region, POR FESR 2014/2020. Nat-BackeryInnov "Innovative production of a bakery line based on natural functional extracts for wellness and sports”; Tuscany Region, POR CREO FESR 2014-2020 “Formulation of new sustainable products for agriculture based on natural extracts”, LIFE18 ENV/NL/000043 - LIFE “PlantsforPlants”.]. 

We note that one or more of the authors are employed by a commercial company: PIN scrl

We would like to thank the Editors for pointing this. The affiliation reported in the manuscript has been completed by adding a lacking part of the description. PIN scrl is not a commercial company, but an University Center for educational and scientific services for the University of Florence, as now indicated (see the affiliations in the Title page). So the funder of this research is not PIN scrl, but Tuscany Region for projects POR FESR 2014/2020 Nat-BackeryInnov "Innovative production of a bakery line based on natural functional extracts for wellness and sports” and POR CREO FESR 2014-2020 “Formulation of new sustainable products for agriculture based on natural extracts”, and the European Community for LIFE Project Plants for Plants, “Boost conventional agriculture’s confidence: new organic bio-stimulants to reduce water, nutrients and pesticide demand project”, LIFE18 ENV/NL/000043. So we can exclude the existence of competing interests.

Reviewer #1: This paper needs to be carefully reorganized with a clearly stated argument spelled out in the opening pages. The authors should work harder on the approach adopted, establish a clear theoretical background to contextualize the analysis and narrow the scope of the analysis to specific aspects. Apart from compiling key studies, a critical approach to the literature is required. There is a need of offering detailed insights regarding the main purpose of the paper and its contribution. The discussions require more structure and there is a need of offering a clear assessment of reviewed literature. The main contributions of the paper should be presented as part of the empirical discussions or critical assessment on the core research outcomes. The main contribution should be emphasised more and the concluding statements should be stronger. The paper presents its points in a rather descriptive manner, referring to relevant sources but without really discussing them. It brings up interesting points, but stating them rather than arguing for them. The research idea is not properly contextualised, as there is a need of offering a detailed review of relevant literature that help the authors developing the key arguments that support their proposed research. Data gathering and data analysis can be reconsidered and discussed more comprehensively. The lead up argument for the study and results section appear to be too loosely constructed. There is some discussion of the limitations of the study however these are not considered in terms of the implications on the study findings. Some bibliographic references are simply brought up without being developed, or without an adequate explanation as to why they are relevant.

We would like to thank the reviewer for his/her comment. As he/she suggested, the paper has been reorganized, in particular the “Introduction” and “Results and Discussion” sections. Both the results and literature were discussed more in depth, also by adding some new references to recent studies, to better contextualize our study and explain the main purpose and contribution of the paper.

Reviewer #2: Reviewer’s Report: PONE-D-20-10008

Line 316-318 figures title should be removed: we would like to thank the reviewer for his/her comment. According to PlosOne Manuscript body formatting guidelines, “Each figure caption should appear directly after the paragraph in which they are first cited”. For this reason we have kept the figure captions in the same position.

Line 353 space should be removed: the space has been removed.

• Furthermore, although no requests were made, we have replaced the file of Figure 1 by uploading a new one of better quality, as we noticed that the previous image had an insufficient quality.

---

## [Decision Letter · Decision Letter 1]

5 Jan 2021

PONE-D-20-10008R1

Sweet chestnut (Castanea sativa Mill.) standardized fractions from sustainable circular process and green tea (Camellia sinensis L.) extract: in vitro inhibitory activity against phytopathogenic fungi for innovative applications in green agriculture

PLOS ONE

Dear Dr. Campo,

Thank you for submitting your manuscript to PLOS ONE. After careful consideration, we feel that it has merit but does not fully meet PLOS ONE’s publication criteria as it currently stands. Therefore, we invite you to submit a revised version of the manuscript that addresses the points raised during the review process.

MS still needs a few corrections before it maybe considered for publication in PLSO One.  Kindly do the needful changes as motioned by the reviewers and submit a revised MS.  

We look forward to receiving your revised manuscript.

Kind regards,

Vijai Gupta, PhD in Microbiology

Academic Editor

PLOS ONE

Additional Editor Comments (if provided):

MS still needs a few corrections before it maybe considered for publication in PLSO One. Kindly do the needful changes as motioned by the reviewers and submit a revised MS.

Reviewers' comments:

Reviewer's Responses to Questions

**Comments to the Author**

1. If the authors have adequately addressed your comments raised in a previous round of review and you feel that this manuscript is now acceptable for publication, you may indicate that here to bypass the “Comments to the Author” section, enter your conflict of interest statement in the “Confidential to Editor” section, and submit your "Accept" recommendation.

Reviewer #3: (No Response)

Reviewer #4: All comments have been addressed

2. Is the manuscript technically sound, and do the data support the conclusions?

Reviewer #3: Partly

Reviewer #4: Yes

3. Has the statistical analysis been performed appropriately and rigorously? 

Reviewer #3: Yes

Reviewer #4: Yes

4. Have the authors made all data underlying the findings in their manuscript fully available?

Reviewer #3: Yes

Reviewer #4: Yes

5. Is the manuscript presented in an intelligible fashion and written in standard English?

Reviewer #3: No

Reviewer #4: Yes

6. Review Comments to the Author

Reviewer #3: (No Response)

Reviewer #4: I have gone through the manuscript and found very informative research work on antimicrobial activities of extract took from tea in inhibitory test against pathogen. I strongly recommending to this research paper for accepting to publish in forthcoming issue of the journal.

7. PLOS authors have the option to publish the peer review history of their article (what does this mean?). If published, this will include your full peer review and any attached files.

Reviewer #3: No

Reviewer #4: **Yes: **I have gone through the manuscript and found very informative research work on antimicrobial activities of extract took from tea in inhibitory test against pathogen. I strongly recommending to this research paper for accepting to publish in forthcoming issue of the journal.

---

## [Author Response · Author response to Decision Letter 1]

20 Jan 2021

Dear PLOS ONE Editors and Reviewers,

All the authors would like to thank the Editor and Reviewers for their careful reading of the manuscript PONE-D-20-10008R1 entitled “Sweet chestnut standardized fractions from sustainable circular process and green tea extract: in vitro inhibitory activity against phytopathogenic fungi for innovative applications in green agriculture”.

The manuscript has been revised according to all Reviewers’ comments and all corrections have been marked-up in the copy labelled “Revised Manuscript with Track Changes”; as requested, also an unmarked version without tracked changes, labelled “Manuscript”, has been uploaded. 

Comment 1:

Author should consider using either common name or scientific name in the title and use both on the first mention in the manuscript. Then choose which to use throughout and make it consistent.

For sweet chestnut and green tea, only common names were kept in the title; on the first mention both in the Abstract and in the text (Introduction section), common names with the complete scientific names are reported; only common names are used for the following mentions. Also pathogens names were checked and fixed.

Abstract: Line 39-40: “can attack also plants in the field” can be stated as “can also attack field plants”. 

Done.

Comment 2:

Line 83: Should include the following reference:

Mohammed Bule et al. Tannins (hydrolysable tannins, condensed tannins, phlorotannins, flavono-ellagitannins). Recent Advances in Natural Products Analysis. Elsevier, 2020, pp. 132-146.

Though the introduction is relevant and theory based, sufficient information on previous findings is not presented well for the readers to follow the present study rationale.

The reference was added in the text. 

The “Introduction” section has been revised and improved (lines 156-167 in particular) in order to let the readers better understand our study rationale. 

Comment 3:

In method section: Lines 167-173: Author may consider including the flow diagram of the dry extraction process.

The flow diagram of the extraction and fractionation industrial process was already reported in a previous publication (Campo et al., 2016), so it is no more possible to use the figure. Here we cited the articles where we described accurately the whole process also by showing the flow diagram (Campo et al., 2016; Lucarini et al., 2018).

Comment 4:

Line 226-227: Must provide the reference to “previously described”. 

As it could seem like a reference to literature, in the place of “as previously described”, we have inserted “as described above (see “In vitro test of radial growth inhibition”)”. The reference, indeed, is to what described in the previous paragraph.

Comment 5:

Line 266: Consider using “Campo et al.” instead of “this research group”. 

“Campo et al.” was added instead of “this research group”.

Comment 6:

Line 275: Please clarify “6-months and 12-months” or “12-months”.

In the place of “The dried SCW fraction was chemically stable at the 6-months and 12-months as demonstrated by the analytical controls”, we added the following sentence: “The dried SCW fraction was chemically stable as demonstrated by control HPLC/DAD analysis repeated after 6 and 12 months.”. This new sentence should be more clear than the first one.

Comment 7:

Consider proofreading the manuscript.

The whole manuscript has been carefully revised and checked. Some typos were fixed and some sentences were re-written to make them more clear. Particular attention was paid to the English language.

---

## [Editor Report · Decision Letter 2]

5 Feb 2021

Sweet chestnut standardized fractions from sustainable circular process and green tea extract: in vitro inhibitory activity against phytopathogenic fungi for innovative applications in green agriculture

PONE-D-20-10008R2

Dear Dr. Campo,

We’re pleased to inform you that your manuscript has been judged scientifically suitable for publication and will be formally accepted for publication once it meets all outstanding technical requirements.

Kind regards,

Vijai Gupta, PhD in Microbiology

Academic Editor

PLOS ONE

Additional Editor Comments (optional):

All the comments have been addressed. Now paper can be accepted for publication in PLOS One.
---

## [Editor Report · Acceptance letter]

12 Feb 2021

PONE-D-20-10008R2 

Sweet chestnut standardized fractions from sustainable circular process and green tea extract: *in vitro* inhibitory activity against phytopathogenic fungi for innovative applications in green agriculture 

Dear Dr. Campo:

I'm pleased to inform you that your manuscript has been deemed suitable for publication in PLOS ONE. Congratulations! Your manuscript is now with our production department. 

Kind regards, 

on behalf of

Dr. Vijai Gupta 

Academic Editor

PLOS ONE